# Triad influence on the detection of crime in Hong Kong

**Gabriel Wong[1‡], Matthew Manning[2‡], T. Wing Lo[2,3‡], Shane D. Johnson[4]***

1 Centre for Social Research and Methods, Australia National University, Canberra, Australia, 2 Department of Social and Behavioural Sciences, City University of Hong Kong, Hong Kong, China, 3 Office of the President, Caritas Institute of Higher Education, Hong Kong, China, 4 Department of Security and Crime Science, University College London, London, United Kingdom

‡ GW, MM and TWL are co first authors on this work.
* shane.johnson@ucl.ac.uk

**Data Availability Statement:** The data analyzed were requested from Hong Kong police. Their provision incurs a cost and, due to their sensitivity, we are unable to share these data ourselves. Individual's wishing to use these data can make a request for access to Hong Kong Police (which will

## Abstract

We use bootstrap data envelopment analysis, adjusting for endogeneity, to examine police efficiency in detecting crime in Hong Kong. We address the following: (i) is there a correlation between the detection of crime and triad influence? (ii) does the level of triad influence affect the efficiency in translating inputs (police strength) into outputs (crime detection)? and (iii) how can the allocation of policing resources be adjusted to improve crime detection? We find that nighty-eight percent of Hong Kong police districts in our sample were found to be inefficient in the detection of crime. Variation was found across districts regarding the detection of violent, property and other crimes. Most inefficiencies and potential improvements in the detection of crime were found in the categories violent and other crimes. We demonstrate how less efficient police districts can modify police resourcing decisions to better detect certain crime types while maintaining current levels of resourcing. Finally, we highlight how the method we outline improves efficiency estimation by adjusting for endogeneity and measuring the conditional efficiency of each district (i.e. the efficiency of crime detection taking the instrumental variables (e.g. influence of triads) into consideration). The use of frontier models to assist in evaluating policing performance can lead to improved efficiency, transparency, and accountability in law enforcement, ultimately resulting in better public safety outcomes and publicly funded resource allocation.

## 1. Introduction

Organized crime groups (OCG) threaten multiple facets of a country through activities such as drug trafficking, human smuggling, money laundering, and corruption [1]. The effects can be experienced directly or indirectly by individuals and society [2]. Individuals may be directly affected because of victimisation or indirectly by resulting harms such as a diminished sense of safety and wellbeing or economic losses. From a societal perspective, harms include: (i) the malignant impact on governance, such as the corruption of officials; (ii) market distortions; (iii) loss of local and foreign investment as a result of poor confidence in the legal and

come at a cost to the requester). Publicly accessible data is available from https://www.police.gov.hk/ppp_en/09_statistics/as.html. The data analysed in the paper differs from the publicly available data in that it is: (1) for a longer period of time than that which is publicly available (2) provided at the district level; (3) supplied at the lowest level of crime type; and (4) it includes arrest data which is not publicly available. Triad data are be made available in the Supporting Information files and are de-identified due to the ethical concern that communities and districts, if identified, may be subject to heightened triad influence and change in safety risks if information on triad influence are disclosed. Sharing such data could potentially expose individuals in specific districts to harm.

**Funding:** The author(s) received no specific funding for this work.

**Competing interests:** The authors have declared that no competing interests exist.

regulatory functions of the state; and (iv) the distortion of public procurement and the health and social costs from crime [3–5].

To reduce the above-mentioned harms, much effort is undertaken by authorities in the prevention, detection and prosecution of organised criminal activities. However, equal effort, if not more (given the profitability of these activities), is most likely undertaken by OCG to make the detection of their crimes difficult. One method is to engage professional facilitators who assess commodities, identify criminal opportunities, and/or specialise in methods of detection avoidance. In the event that their activities are detected, OCG utilize a number of strategies to make the offence harder to prove, for example: (i) using traditional methods of intimidation and concealment; and (ii) influencing criminal justice personnel using strategies such as bribery or blackmail [6].

To mitigate the strategies adopted by OCG to conceal their activities, the effectiveness of policing is dependent upon a combination of police management skill, tactics, good police-citizen relationships, investigative ability and relevant knowledge across the entire investigative process–from initial crime scene assessment through to post-charge case management [7–9]. Typical approaches used by police to enhance the efficacy of crime detection include examining and modifying police strength [10], police composition [11], and method of policing (e.g. foot patrols, undercover policing) [12, 13]. The implicit assumption is that a positive change in any of these approaches can lead to a higher probability of identifying and detaining offenders [14].

To date, few studies have explored whether the efficiency of the detection of crime can be improved by modifying any of the above variables such as police strength or composition (e.g. detectives and frontline police). Further, when inefficiencies are identified, how can police strength be redirected to enhance efficiencies in low detection districts?

In this paper, we attempt to answer the above question using data from Hong Kong that include information about the activities of triad gangs, who arguably work hard to conceal their illegal activities from police. Above, we described how OCGs can indirectly affect detection by influencing the way police deploy resources. Typically, one would expect that districts with a higher triad influence would have a lower detection of crime. However, this has not been empirically tested. Our thesis is that the choice of police skills and where these skills are distributed is critical to the disruption of organized crime activities. We attempt to address the following research questions:

1. *Are there correlations between triad influence, the detection of crime and the number of police in Hong Kong at a district level?*

2. *Does the efficiency in translating inputs (i.e. human resources) to outputs (i.e. the detection of crime) differ across districts that display different levels of triad influence?*

3. *Which districts are efficient in the detection of crime and how can inefficient districts modify their allocation of policing resources to enhance efficiency in the detection of crime?*

An understanding of these points will help law enforcement and policymakers to develop more effective and efficient policing strategies to detect criminal activity at the district level.

The paper proceeds as follows–first, we introduce our study site by reviewing a brief history of policing strategies in Hong Kong as well as the origin, structure and operation of triad society, providing an explanation of the recent transformation of triad society in Hong Kong, and by illustrating how triads influence the detection of crime. Next, we discuss how police detect crime as well as the variables used to measure the efficiency of policing services directed at detecting crime. We then present results including efficiency scores for the 18 Hong Kong police districts and information on where resources can be modified to enhance the efficiency

of detection. While the scenario examined in this paper is particular to Hong Kong, the approach taken and the findings presented have much wider applicability.

## 1.1 Organised crime in Hong Kong

**1.1.1 A brief history of Hong Kong policing strategies.** Hong Kong society and policing has undergone transformations because of changing community expectations and institutional responses to the evolving nature of crime in the region. Shifts in policing strategies and approaches can be seen through various eras. This section provides a brief overview of some recent key transformations.

The origins of modern policing in Hong Kong can be traced to the colonial era (mid-19th Century to 1997), when Hong Kong was still under British colonial rule. In this era (from mid-19th Century to early 1990s), policing was characterised by the para-military structure and policing model, focusing on the maintenance of law and order, particularly in the densely populated urban areas [15]. In the early 1990s, Hong Kong police Force began to move away from a para-military policing model to a more service-oriented community-focused modern police force [16]. This change may be attributed to rising expectations of quality service from the public sector due to an increase in citizens' education level and improvements in the standard of living. Upon the transition of Hong Kong to Chinese Sovereignty in 1997, there was a need for continuity in policing while still addressing the unique challenges of the newly defined Special Administrative Region. Here, the principle of "one country, two systems" was also applied to policing, preserving some of the colonial-era police structure and practices.

In response to global challenges such as terrorism, transnational organised crimes, and cyber and technology crimes, Hong Kong police recognised the need for further change [17]. They saw the need to co-operate beyond Hong Kong geographical boundaries and begin forging partnerships with key law enforcement agencies worldwide. During this time, Hong Kong police began adopting new technologies to respond to new forms of crime that were being assisted and facilitated by information and communication technology. The Hong Kong police [18] identifies 8 Force values that underpin the discharge of their duties: (1) integrity and honesty; (2) respect for the rights of members of the public and of the Force; (3) fairness, impartiality and compassion in all dealings; (4) acceptance of responsibility and accountability; (5) professionalism; (6) dedication to quality service and continuous improvement; (7) responsiveness to change; and (8) effective communication within and outwith the Force. All the above are at the core of Hong Kong police efforts to embed integrity and an ethical culture in all aspects of policing. It is important to note that the dynamics of policing in Hong Kong are influenced by a complex mix of historical, political, and social factors. These all have significant impact on the strategies employed and the priorities identified to deal with modern crimes in Hong Kong societies.

**1.1.2 The origin, structure, and territoriality of triad society.** Triad society, the most renown form of OCG in Hong Kong, originated from the Hung Mun which was founded in 17th Century China [19]. The Hung Mun had a strong patriotic doctrine to overthrow the Qin Dynasty to restore the Ming Dynasty. Post-World War II, triad society began to fragment into several independent societies that changed their nature from a patriotic secret society with mutual assistance groups into purely criminal syndicates [20]. Somewhat different from most other OCGs, triad society is typically based on the fictive-kinship affiliation through formal rituals, where membership is neither consanguineal (i.e. blood ties) nor affinal (i.e. by marriage) [21]. Triad members are bound by triad subculture, which includes sharing a common identity, involvement in triad rituals and strict adherence to group values such as brotherhood and loyalty [22]. Traditionally, triad society is perceived as a single entity with a hierarchical and

cohesive structure. In reality, its structure is vast and complex, which includes different consortia, each consisting of many independent triad societies or groups. Each triad society is disaggregated into different factions which are led by senior triad members operating in different triad-controlled territories.

Currently, *14K*, *Sun Yee On* ('新義安') and *Wo Shing Wo* ('和勝和') are the largest triad societies in Hong Kong [23]. Disregarding any triad-specific differences, a triad society, in general, adopts a three-tier structure. The top tier–the headquarters–consists of a *Cho Kun* ('坐館', the commander or CEO in corporate terms), *Cha So* ('揸數', Treasurer), *Heung Chu* ('香主')–who is responsible for triad ceremony and rituals–and a board of *Lo Shuk Fu* ('老叔父', board of directors in corporate terms) who are responsible for promotion, arbitration and the nomination and election of the *Cho Kun* [20]. The second tier comprises of territorial triad bosses or faction leaders who are responsible for operational activities at the district level. The bottom tier includes ordinary and probationary members who are known as "holding the blue lantern" [22].

While it is generally believed that all triad activities are centrally organized by *Cho Kun*, most triad activities are in fact operated by territorial triad leaders [20, 24]. Territory is important to triad society as they need to rely on territory to maintain and establish networks, source business opportunities, and exchange market information [22]. Triad societies tend to operate within their own turf, or monopolise certain geographical territories or economic sectors [23]. As such, most of their legal and illegal activities take place within these controlled territories.

**1.1.3 The transformation and theory underlying triad operations.** A series of criminal justice reforms in Hong Kong in the late 2000s empowered law enforcement agencies and strengthened legislation against organized crime. This led to a stable decline in triad activity and a transformation in triad organizational operations [19, 25]. Specifically, triad activities expanded beyond conventional triad businesses (e.g. prostitution, gambling, and drug dealing [19], extortion and provision of protection services [20, 25]) to quasi-legitimate businesses within legal grey areas. Examples of these businesses include casinos [26], mini-bus franchises, mahjong parlors, film production and dissemination, and waste disposal and recycling [20] as well as quasi-legal financial services and products (e.g. manufactured fraudulent business information to obtain financial gain through insider trading) [27]. In addition, triad societies increasingly expanded their operations across borders exploring new opportunities in overseas markets including China [19].

Lo and Kwok [26] articulate a rationale behind the recent expansion in activities suggesting that traditional operations, which involve the use of violence, negatively impacts triad business. The traditional triad model attracts attention from law enforcement to both triads and their business partners. Some triad business partners, for example, are less likely to obtain greater profit and market power using illegal means [19]. However, it should be recognized that traditional crimes still remain in the remit of OCG and, as such, the expansion into other areas (legal/illegal) represents diversification for the purposes of growth and maintenance of power and influence [28]. The relationship between legal and illegal markets is what von Lampe [29] recognises as the legal-illegal nexus.

Enterprise theory, as introduced by Smith [30] for the understanding of OCG operations, suggests that organized crime exists as a response to market demand and an unsatisfactory supply of a particular good or service (e.g. drugs, prostitution, arms, slavery). Later, Smith's [31] spectrum-based theory of enterprise suggested that OCGs proliferate in environments where there is a low level of risk of detection and potential high profits associated with the supply of those goods and services. These studies provide a rationale for triads recent involvement in complex profitable legal and illegal activities.

## 1.2. Improving crime detection

Improving the detection of crime requires an understanding of the mechanisms of crime detection. In the context of this study, we define detection as police making an arrest as a result of sufficient evidence to indict. Police use a combination of practices, procedures, processes, routines, conventions, theories and techniques to inform the methods (e.g. searches, use of informants, foot patrol locations) for acquiring and interpreting information that is useful for detecting different crimes and adapting to the different conditions in which crime occurs [32].

Effective implementation of police procedures to gathering intelligence and detecting crime depends on a number of activities including developing or maintaining police-citizen relationships, enhancing police legitimacy [33], investigating how criminals operate (e.g. *modus operandi*) and the use of technologies to evade (if on the criminal side) or detect (if on the criminal justice side) crime. As such, police time spent on policing activities (e.g. what the typical uniformed officer does in a shift) is most likely unequally distributed among the various groups that make up the police force.

The fraction of police resources attributed to targeting crime in triad-influenced districts is currently unknown. However, we do know how police strength (in terms of uniformed police and detectives) is distributed across the 18 Hong Kong police districts. Since triad activities involve both territorial-based street or youth gangs [34] and entrepreneurs or 'racketeers' [26], the deployment of police strength including both uniformed police and detectives is necessary to detect these activities. In addition, since triads disproportionately occupy districts, their influence on individual communities and the policing procedures used to detect crime within them will most likely vary. For example, some of the lucrative commercial night-time entertainment districts (e.g. Mongkok, Yao Ma Tei) have historically been dominated by local triads which run the brothels, karaoke bars, snooker halls, and red-light district premises [35, 36].

Based on the above knowledge, we propose that much insight regarding detection can be gained by coupling police strength data with district-level triad influence data. Doing so allows us to use an empirical method (frontier modelling) to reveal where improvements in detection can be made, at the district level, via police deployment. In other words, how outcomes (i.e. detections) can be improved using current policing resources. Several studies have been conducted using frontier modelling in jurisdictions including for example, Australia [37], Belgium [38], England and Wales [39], Israel [40], Slovenia [41, 42], the US [43], Spain [44], Taiwan [45], Punjab [46], Mexico [47], Peru [48] and Lahore [49]. A list of these studies with information regarding their model and selection of variables is available in S1 Appendix. However, this is the first study to combine policing data with triad influence data and thus represents a leap forward in our ability to assist police in undermining the nefarious triad-related activities. It is important to note that DEA or frontier methods are not the only available resource to identity criminal practice and their influence on police activities. For example, text mining techniques and latent Dirichlet allocation can be adopted to obtain topics related to crime from police response service database (e.g. police occurrence reports) to construct strategies for responding to crime at the local and regional level [50]. In this paper, we use DEA as the guiding method for performance evaluation as it aligns with previous evaluations in this context [51], contributing also to the field of instrument-adjusted DEA for law enforcement and other public sectors as proposed by Santin and Sicilia [52].

## 2. Data

### 2.1 Administrative data

We use administrative crime data (from 2007 to 2017) derived from the Hong Kong Police Force (Force from hereon) Annual Digest [53]. The data are available at the district level by crime type for the categories violent (e.g. murder, wounding), property (e.g. shoplifting, burglary) and crimes not classified as violent or property crime (e.g. drug offences, fraud, sexual offences). We sampled 18 police districts in Hong Kong. In 2017 these districts had a mean population of 406,265 (SD = 205,447). Similar to the UK system, there are four main principles adopted for counting crimes: (1) Violent crime against persons–here, one crime is counted for each person against whom an offence has been committed. For example, a gang attack involving four victims, where there is one killing and three injuries, would be counted as one offence of murder and three offences of wounding; (2) Crime against property or public order–crime is counted for each distinct offence. For example, a case where two men armed with pistols who rob a bank with 10 customers inside would be counted as one offence of robbery; (3) Crime considered as a continuous offence if repeated (e.g. crimes such as blackmail and deception). Here, one crime would be counted for each offence or series of similar offences committed by the same offender or group of offenders on different occasions involving the same victim or group of victims. For example, if three offenders blackmailed a shop-owner multiple times in the past three months this would be counted as one offence of blackmail; and, (4) Crime associated with other naturally connected offences–here, when multiple related criminal acts are performed, the official statistics would capture the most serious offence as the principle offence. For example, if an offender is arrested for attempting to steal from a vehicle which involves damage to property, the offence of vehicle theft would be counted rather than offence of criminal damage or offence of possession of instruments for unlawful purposes.

The period 2007–2017 was selected to avoid known distortions in crime statistics because of unusual police practices employed during social movements (Anti-Extradition Law Amendment Bill Movement) and COVID-19 lockdown restrictions as well as changed offending patterns during, as well as the months prior to, these periods.

### 2.2 Primary survey data

Altogether 200 male incarcerated respondents from eight correctional institutions, covering minimum to maximum security level, were invited to participate in the study. After giving informed consent, they were organised into small groups to fill in a self-administered questionnaire delivered by a research assistant. They signed a written consent form, which explained the study's academic purpose, confidentiality and anonymity, and their right to withdraw from the study anytime during the process if they felt uncomfortable. The names of respondents were not revealed to the research group. Each respondent was coded as a number with no identifying name or prison Id number. No photos were taken of respondents as the act breaches prison rules. It is therefore not possible to identify any respondents.

The study was conducted between 1st December 2017 and 31th January 2018. It adopted a purposive sampling method, and only individuals who self-identified as current or former members of a triad organisation were selected. The sample distribution in eight Hong Kong correctional institutions is presented in Table 1. Respondents were invited to answer a question on the seriousness of triad influence in the 18 police districts of Hong Kong. The question was: "According to your observation, please rate the seriousness of triad influence in the following districts using a scale of 1 to 10, where 1 means the least serious; 10 means the most serious."

**Table 1. Distribution of respondents from each correctional institution.**

| Correctional institution | N | % |
|:---:|:---:|:---:|
| A | 66 | 33 |
| B | 34 | 17 |
| C | 22 | 11 |
| D | 20 | 10 |
| E | 18 | 9 |
| F | 14 | 7 |
| G | 14 | 7 |
| H | 12 | 6 |
| Total | 200 | 100 |

## 3. Method

We employ the data envelopment analysis (DEA) method to examine how policing inputs (i.e. human resources) can be modified to improve the detection of crime across the 18 police districts in Hong Kong. DEA is a frontier modelling method that utilises a mathematical programming approach for constructing a production possibility frontier (PPF). The PPF measures the relative efficiency of similar organisational entities (here, different police districts) in utilising inputs (e.g. police strength) to generate improvements in outputs (e.g. the detection of crime). The PPF can be used to: i) reveal the combinations of outputs that could theoretically be generated using the same fixed total amount of each input; or ii) identify how inputs can be modified to produce outputs that are consistent with the 'most' efficient organisational entities. In this paper, we examine how police districts, defined as decision-making units (DMUs), can theoretically modify police strength and tactics to be as efficient as a district identified as efficient based upon the PPF.

The slope of the PPF, at any point, as shown in Fig 1 is called the marginal rate of transformation (MRT). The slope expresses the rate at which production of one good or service can be redirected into production of the some other good. Economists call this the (marginal) 'opportunity cost' of a good. That is, the opportunity cost of $good_H$ in terms of $good_M$ at the margin. The PPF measures how much of $good_H$ is given up for one more unit of $good_M$ or vice versa. The shape of the PPF is commonly concave to the origin to represent the increasing opportunity cost with increased output. Therefore, the MRT increases in absolute size as there is movement from the top left of the PPF to the bottom right [54]. In simple terms, we illustrate a PPF as the opportunity cost of a decision between two alternative choices; for example, two different outputs in detecting two types of crime (see Fig 1). The dots labelled A, B, X and Z show the combinations that can be made with respect to two fictitious alternatives (1 and 2). Any points on the PPF line (Points A and B) are said to be efficient and an indication that scarce policing resources are being fully employed. Any point inside the PPF (e.g. point X) is said to be inefficient because output could be produced more efficiently using existing resources or factors of production. Any point outside the PPF (e.g. point Z) is impossible given current resource limitations. Only increases in technology will allow this point to be reached–this would be shown as an outward shift in the PPF (dotted curve line). The PPF also assists in making decisions regarding whether to specialise or how changes can be made to improve the efficiency of an outcome such as X in Fig 1. In our study, X could be a police district which underperforms in the detection of some crimes when compared to the most efficient districts (i.e. those sitting on the PPF). The DEA process would allow us to identify what factors (e.g.

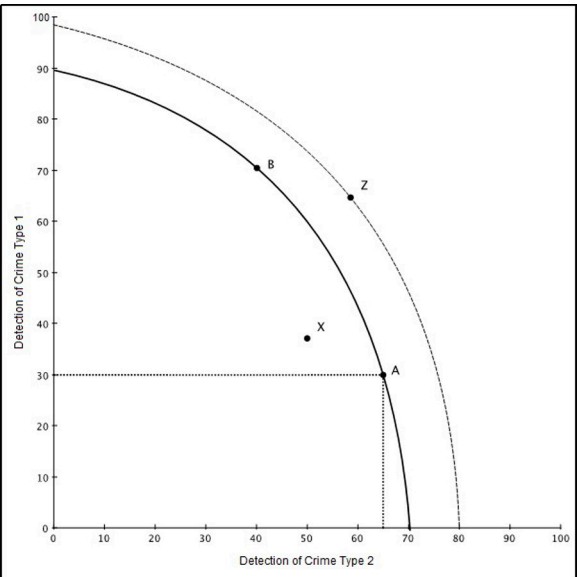

**Fig 1. Illustration of the PPF.**

policing resources) can be modified so that X falls onto the frontier similar to A and B, which in this case would be the most efficient districts.

While Fig 1 presents a simple scenario with only two outputs (detection of crime types 1 and 2) and one input (generic police resources), most DEA models cannot be illustrated in this way as they include more inputs and outputs. But this is common and is the case with our model. Details are provided below in the method section.

There is often confusion between regression models and the DEA technique. In a regression analysis, one would be interested in how some independent variables (e.g. the number of police, the population) potentially influence the detection of crime–the dependent variable, in the aggregate sense. This technique, however, does not address the DMU-specific institutional decision-making in being more efficient in their allocation of resources so that individual DMUs can transition to points on the PPF frontier (such as points A and B in Fig 1). Regression, as a parametric method, requires the specification of a general model examining the relationship between input and output levels. DEA, however, is a non-parametric 'boundary' method which is not limited to the specification of a single set of parameters and applies the PPF to identify the distance between the benchmark DMUs (i.e. DMUs in Fig 1 that sit on the PPF such as A and B) and the less efficient DMUs [55]. Hence, DEA, rather than suggesting an association between the number of police and the detection of crime, focuses not on the aggregate but individual level processes that can be modified to improve individual DMU efficiency. Readers who are interested in an in-depth explanation of the concepts and complex methodological foundations of the PPF and DEA models should consult our previous publication [56].

## 3.1 Important methodological considerations

Basic DEA models require a choice of input/output orientation and an assumption on scale of inputs/outputs conversion (i.e. constant returns-to-scale (CRS) or variable returns-to-scale (VRS)). The VRS can be understood as either increasing returns-to-scale (IRS)–one unit of input results in more than one unit of output, or decreasing returns-to-scale (DRS)–one unit of input results in less than one unit of output. The original DEA-CCR model [57] assumes a

CRS. The model may be input-oriented, which attempts to minimise inputs while adequately satisfying the given output level, or output-oriented, which attempts to maximise outputs with a given input level. Most frontier models for policing efficiency adopt an output-oriented approach (see S1 Appendix).

With $n$ DMUs each having $m$ inputs and $r$ outputs as defined by the vectors $x_i = (x_{1i}, x_{2i}, \cdots, x_{mi})^T \in R^m$ and $y_i = (y_{1i}, y_{2i}, \cdots, y_{ri})^T \in R^r$, featuring the input matrix $X$, defined as $X = (x_1, \cdots, x_m) \in R^{m \times n}$, and the output matrix $Y$, defined as $Y = (y_1, \cdots, y_r) \in R^{r \times n}$ and assuming that $X > 0$ and $Y > 0$. The output-oriented CCR model can be defined as following linear programming. For $DMU_p$, we have the following:

$$\rho_p^* = max\rho_p \tag{1}$$

$$subject\ to\ \sum_{i=1}^{n} \alpha_i x_{ji} \leq x_{jp},\ (j = 1, 2, 3, \ldots, m)$$

$$\sum_{i=1}^{n} \alpha_i y_{ki} \geq \rho_p y_{kp'}\ (k = 1, 2, 3, \ldots, r)$$

$$\alpha_i \geq 0.$$

When using DEA to assess the efficiency of police or other law enforcement agencies in a group of DMUs, the linear programming algorithm computes the efficiency of each DMU. This computation is based on the same input and output variables, aiming to identify the DMU that achieves the highest ratio of the weighted sum of outputs to the weighted sum of inputs. The DMU with the highest ratio serves as a benchmark against which other DMUs are compared. Consequently, the most efficient DMUs are positioned along the efficiency frontier line. In practical terms, this means that the DMUs considered as best practice are relatively efficient, as indicated by their DEA efficiency score, which is typically 100% (efficiency = 1).

With an alternative set of assumptions, the DEA-BCC model [58] adopts the VRS allowing the generation of DMU-specific scale-efficiency information for each DMU on the PPF. The BCC model introduces an extra constant variable to accommodate VRS, allowing for the possibility that the outputs may change at a different rate compared to the inputs. This adjustment addresses the concern related to the size of the units under analysis. The following constraint is imposed on the Eq (1).

$$\sum_{i=1}^{n} \alpha_i = 1. \tag{2}$$

Numerous other adaptations are readily expandable and can be found in the majority of DEA literature, as exemplified by authors such as Coelli, Rao, O'Donnell, and Battese [59], as well as Cooper, Seiford, and Tone [60].

In both IRS and DRS, the translation of inputs and outputs could be a relative change. Such information allows the most productive scale size and efficient DMU to be identified relative to other DMUs. DEA measures weighted output to weighted inputs and thus the efficiency score ranges from 0 to 100 (as a ratio of the inputs and outputs), where the value of 100 means efficient and less than 100 means inefficient. The overall technical efficiency (commonly known as the *overall efficiency* measured under the DEA-CCR model) can be decomposed into pure technical efficiency (commonly known as *technical efficiency* measured under the DEA-BCC model) and scale efficiency (i.e. overall efficiency divided by technical efficiency).

In other words, a DMU is deemed to be scale efficient if it has the same efficiency score from both DEA-CCR and DEA-BCC models.

## 3.2 Our sample

We examine crime detection efficiencies across 18 police districts for a period of 11 years. This requires an assumption that it is reasonable to use a window analysis (pooled frontier analysis), so a unit can be compared to units from other periods. A window analysis assists in analysing trends and potential stability in crime detection efficiency for each police district [61]. Each district is, therefore, represented as if it were a different unit of analysis for each of the 11 successive years in the window (2007–2017). This results in an analysis of 198 (18 x 11) DMUs. Similar to Sun [62], we employ a window analysis across all years in our reference period and then consider the average efficiency across the years for each district and the average efficiency score in each year across districts. We analysed the productivity changes in efficiency over time using the Malmquist productivity index (MPI) [40]. We found very little discrimination in our models. That is, we could only identify very little inefficiency, and this is potentially the result of dimensionality (especially in VRS) rather than the inefficiency of police districts in Hong Kong. Hence, we continue to employ window analysis.

Eighteen out of 21 police districts were selected as they provide services to approximately 98 percent of the population and account for more than 95 percent of all recorded crimes in Hong Kong. The three districts excluded were the Marine region, and the airport and border districts. These were excluded because the organisation, operation and style of policing are different to regular civil policing in other policing divisions [63].

The number of units of analysis (i.e. 198 DMUs across the reference period) included in our study satisfies the recommended minimum proposed by Friedman and Sinuany-Stern [64] and Cooper, Seiford [65] that $n$ should be greater than the total of three times the number of input and output variables ($n \geq 3*(2+3) \geq 15$).

## 4. Procedure

### 4.1 Step 1: Method used to answer research question 1

We employ Pearson Correlation to determine whether triad influence, the detection of crime and the number of police (detectives and uniformed police) are correlated.

### 4.2 Step 2: Method used to answer research question 2

**4.2.1 Inputs and outputs for our efficiency model.** Our model has two inputs, the number of detectives and the number of uniformed police in a district, and three outputs, the number of detections (i.e. arrest) for violent crimes, property crimes and other crimes. The decision of using detected crimes rather than reported crimes (i.e. number of crimes coming to the attention of police but not necessarily leading to arrest) or the ratio of detected-to-reported crime as output is because: (i) the focus of this model is on police efficiency in making arrests based on their finite human resources; (ii) our previously published work [51] demonstrates that a higher crime rate in a district does not lead to a higher DEA efficiency; and (iii) when we factor in the number of reported crimes in a district and the proportion of crimes that were detected, the approach makes little difference to the efficiency results (i.e. through comparing models that use only detected crimes as outputs vs. the ratio of detected-and-reported crimes as outputs).

In this study, we use what is called an output-oriented DEA model. The model focuses on how much output can be proportionally increased without making substantial changes to

inputs. This model is important to use in this context because police resources cannot be easily modified on a large scale in a short period of time. Rather, what often occurs is that police resources are redistributed to where they are most needed. Some DEA studies incorporate regional factors (e.g. population size, geographical size of a given area), however, such factors are not appropriate in this context as they do not contribute directly to the input-output translation. Rather, they form the structure or context in which police operate. These factors are not modified by police to generate improvements in crime detection and, moreover, the influence of population size will likely be captured in the case/workload of officers in a DMU.

**4.2.2 Theoretical justification for our instrumental variables.** Endogeneity (i.e. independent variables are influenced by dependent variables and vice-versa) is frequently observed in economic production processes, but tends to be overlooked in the application of frontier modelling. Without capturing the endogeneity in the inputs, a DEA model risks using biased inputs which could potentially be under- or over-estimating the impact of the inputs drawing suboptimal inferences from the model [66]. Following Simar and colleagues [67], we apply a statistical heuristic procedure that enables the identification of the presence of endogeneity.

If inputs are exogenous, the expected correlation coefficient between the inputs and the DEA estimated efficiency scores $\theta$ for the 198 DMUs, $n = 1, 2, \ldots, N$, should be close to zero. Therefore, the statistical method to identify endogeneity is based on comparing these expected correlation coefficients under the assumption that a context is exogenous or endogenous to classify each input type included in the DEA model. In the event that endogeneity is identified, we employ the use of an instrumental (input) variable (IV) to enable us to measure the conditional efficiency of each district (i.e. the efficiency of crime detection taking the IV (e.g. influence of triads) into consideration).

To adjust endogenous inputs in our DEA model we adopt a semi-parametric strategy that utilizes the well-known IV approach [see, for example, 68, 69] into the DEA model specification. This approach, as proposed by Santin and Sicilia [52], shares the same intuitive idea as the IV strategy, whereby the DEA specification includes only the exogenous part of the endogenous input (i.e. the part that is uncorrelated with technical efficiency). We use OLS in the context below for the purpose of illustration.

For the uninitiated reader, an IV is a third variable, Z, used in regression when endogenous variables are employed. Using an IV to identify the hidden (unobserved) correlation allows the model to capture the true correlation between the explanatory variable (X) and outcome variable (Y). Here, assume that there are two correlated variables that we want to regress: X and Y. Z is correlated with the explanatory variable (X) and uncorrelated with the error term ($\varepsilon$). The correlation between X and Y might be described by a third variable Z, which is associated with X in some way. Z is also associated with Y but only through Y's direct association with X. For example, the correlation between detection of crime (X) and deployment of police officers to detect crime (Y) may be affected by triad influence (Z).

Bushway and Apel [70] state "the key assumption of instrumental variables is that the *only* way the instrumental variable [in our case, triad influence] is correlated with the dependent variable [detection of crime] is through the independent variable [police strength]". We find no convincing evidence in the literature that triads directly influence the detection of crime across districts, but they arguably create more demand on police time, and lead to the allocation of more resources (i.e. police and non-human resources) and skilled police officers being deployed to a given area. Skilled officers are deployed to districts where their skills and expertise assist in the detection of crime particularly where their expertise relate to the problem at hand [71].

More formally, we propose that triad influence (Z1) is correlated with detection of total crime (Y) but only through its relationship with police detectives (X1). Other variables most

likely exist, for example, the demographics of the district population or the proportion of different types of properties (e.g. commercial and residential), that can influence both crime detection and police strength, but crime detection is not directly caused by triad influence. In addition, the size of the population in a given district, which potentially serves as a proxy of community demand for policing, is likely to be a useful instrument for the number of uniformed police. This is because the larger the population the more the demand for uniformed police, whose activity includes, amongst other things, patrolling areas, maintaining order, and responding to callouts on demand. As above, we propose that the perceived influence of triad societies (Z1) and the size of the population (Z2) are correlated with the detection of crime (Y) but only through their relationship with the deployment of uniformed officers (X2).

In our model, the other exogenous variables include the year of measurement and the proportion of young people in a district. The year of measurement is included to capture temporal variations in crime patterns resulting from technological advancements employed by police (e.g. new surveillance equipment) and other factors that change over time. The proportion of young people (aged 15–24) is included as individuals are most crime-prone at young ages and less so when growing older [72]. Young people are the target of triad leaders who recruit them to strengthen their camp of foot soldiers [34]. They are recruited into triad youth gangs as proteges, and they learn subcultural values and crime knowledge and techniques in their daily hangout.

**4.2.3 Statistical justification for our instrumental variables.** We employ steps that mimic the standard 2-stage IV approach (i.e. 2SLS regression) as follows. First, we regress the endogenous input ($X^e$) over the potential instrument(s) and the rest of the exogenous inputs using an OLS model (see equation below),

$$X^e = \alpha + \beta_1 x_1 + \cdots + \beta_{k-1} x_{k-1} + \tau Z + \varepsilon \qquad (3)$$

where $x_{k-1}$ are the $k − 1$ exogenous inputs, Z is the instrumental input and $\varepsilon$ is a random white noise component. Second, we replace the endogenous input by the predicted variable $\widehat{X^e}$ from the OLS model in the previous step and run a standard DEA linear program using the adjusted inputs.

Several diagnostic tests are undertaken to justify the model specification. First, we undertook tests of endogeneity (H0: variables are exogenous). For the number of detectives, the significant Durbin $\chi^2$ (1) = 21.95, $p < 0.001$ and Wu-Hausman $F$ (1,192) = 23.94, $p < 0.001$, suggest that endogeneity was biasing the estimate of the effect of police detectives on the detection of crime (Y). Second, the first-stage regression summary statistics reveal a minimum eigenvalue of 229.84 that is higher than the critical value of 16.38 for the 2SLS size of nominal 5% Wald test at the 10% level, suggesting the chosen instrument (Z1: triad influence) is strong. Turning to the set of statistic diagnostics of the model for the number of uniformed police, the tests of endogeneity were significant Durbin $\chi^2$(1) = 60.92, $p < 0.001$ and Wu-Hausman $F$ (1,193) = 85.77, $p < 0.001$, and the minimum eigenvalue of 25.36 is higher than the critical value of 19.93 for the 2SLS size of nominal 5% Wald test at the 10% level, suggesting that the chosen instruments (Z1: triad influence, and Z2: size of population) are strong. In summary, both the diagnostics statistics and the theoretical foundation explained above support the use of both instruments in our model.

**4.2.4 DEA based on instrument-adjusted inputs.** Based on the above diagnostic tests and theoretical justification we conducted a DEA with instrument-adjusted inputs to answer the question "*does the efficiency in translating inputs (i.e. human resources) to outputs (i.e. the detection of crime) differ across districts that display different levels of triad influence?*" To do

this we replace the endogenous input by the estimated exogenous variable in the input matrix and run a standard DEA linear program as described by Santin and Sicilia [52].

**4.2.5 Bootstrapping DEA.** DEA being a deterministic method does not explicitly consider random error and overall deviation from the PPF. Essentially, this means that DEA estimates may be affected by sampling variations. Few DEA studies account for potential sampling variations. However, in this study we employ bootstrapping to enhance the reliability of the estimates. Bootstrapping is a statistical method for inspecting the accuracy of estimates. The process of bootstrapping involves replicating the sample by mimicking the data generation process to repeatedly estimate parameters. As defined by Wicklin [73],

> [a]n alternative bootstrap technique is called the smooth bootstrap. In the smooth bootstrap you add a small amount of random noise to each observation that is selected during the resampling process. This is equivalent to sampling from a kernel density estimate, rather than from the empirical density (p.1).

A summary of the bootstrap DEA procedure used here is provided by Simar and Wilson [74].

### 4.3 Step 3: Method used to answer research question 3

To answer research question 3, "*Which districts are efficient in the detection of crime and how can inefficient districts modify their allocation of policing resources (i.e. strength and composition) to enhance efficiency in the detection of crime?*", we undertake what is called a slack analysis. The slack analysis of DEA demonstrates in what areas and by how much an inefficient police district could improve their allocation of resources (i.e. inputs) to become as efficient as their benchmark peers. Theoretically, slacks represent the leftover portions of inefficiencies considering the proportional reductions in inputs or increases in outputs [75].

### 4.4 Statistical packages

This study uses StataSE15 [76] for the descriptive statistics, correlation, regression and the 2SLS. All DEA analyses were conducted using the 'Benchmarking' package of the R software. Further details on the 'Benchmarking' methodologies can be found in Banker and colleagues [58].

## 5. Results

### 5.1 Descriptive statistics

Table 2 provides a summary of the average number of detections of violent, property and other crimes as well as the number of detectives and uniformed police officers in the 18 districts over the study period. Notably, the district "Yuen Long" has by far the highest number of crimes and police, both detectives and uniformed, compared to other districts. In contrast, Lantau, being the smallest populated district, has the lowest number of crimes across all three crime categories and the lowest number of police. The relationships between these key variables are examined below.

### 5.2 Research question 1

Results of our correlation analysis (Table 3) reveal that perceived triad influence is positively correlated with the number of detectives and uniformed police, as well as the detection of violent, property and other crimes. As discussed above, triad influence as the IV correlates with the detection of crime but only through the input variables selected in our model. Districts with a greater number of police, both uniformed and detective, have a higher number of

**Table 2. Descriptive statistics of the 18 police districts (2007–2017).**

| Police district | | No. of detected violent crime | No. of detected property crime | No. of detected other crime | Number of detectives | Number of uniformed police officers |
|---|---|---|---|---|---|---|
| Hong Kong Island Region | | | | | | |
| **Central** | Mean (M) | 229.18 | 363.18 | 417.45 | 106.00 | 673.73 |
| | Standard deviation (SD) | 23.57 | 38.05 | 62.13 | 6.28 | 11.23 |
| **Eastern** | M | 386.55 | 787.45 | 488.09 | 132.36 | 616.82 |
| | SD | 103.98 | 91.38 | 67.70 | 0.50 | 1.89 |
| **Wanchai** | M | 338.64 | 770.18 | 549.91 | 135.36 | 533.36 |
| | SD | 66.58 | 63.12 | 60.91 | 0.92 | 1.21 |
| **Western** | M | 329.09 | 445.55 | 381.55 | 111.55 | 594.09 |
| | SD | 59.50 | 64.71 | 78.49 | 3.91 | 3.30 |
| Kowloon East Region | | | | | | |
| **Kwun Tong** | M | 618.00 | 905.27 | 620.18 | 151.09 | 597.55 |
| | SD | 147.91 | 103.85 | 35.44 | 23.72 | 42.85 |
| **Sau Mau Ping** | M | 384.82 | 505.00 | 453.27 | 107.55 | 491.64 |
| | SD | 88.67 | 65.43 | 60.40 | 4.23 | 13.28 |
| **Wong Tai Sin** | M | 443.64 | 570.18 | 544.64 | 132.36 | 658.45 |
| | SD | 85.19 | 86.31 | 81.70 | 0.50 | 3.53 |
| Kowloon West Region | | | | | | |
| **Kowloon City** | M | 380.55 | 567.73 | 531.18 | 131.09 | 673.45 |
| | SD | 53.06 | 77.92 | 84.48 | 2.47 | 1.37 |
| **Mongkok** | M | 422.27 | 945.55 | 1122.00 | 181.00 | 501.00 |
| | SD | 68.08 | 82.93 | 178.42 | 3.90 | 13.96 |
| **Sham Shui Po** | M | 544.09 | 849.09 | 856.00 | 164.91 | 698.82 |
| | SD | 73.33 | 78.17 | 108.04 | 1.51 | 3.63 |
| **Yau Tsim** | M | 478.91 | 815.09 | 997.27 | 195.82 | 743.64 |
| | SD | 77.89 | 90.18 | 178.50 | 2.52 | 3.26 |
| New Territories North Region | | | | | | |
| **Tai Po** | M | 508.18 | 810.36 | 704.00 | 174.91 | 588.73 |
| | SD | 70.46 | 86.44 | 70.43 | 4.04 | 3.82 |
| **Tuen Mun** | M | 504.09 | 786.36 | 597.91 | 154.36 | 544.00 |
| | SD | 125.58 | 85.74 | 106.30 | 0.50 | 1.79 |
| **Yuen Long** | M | 745.55 | 979.91 | 933.55 | 193.36 | 780.73 |
| | SD | 146.86 | 80.42 | 108.35 | 0.50 | 15.75 |
| New Territories South Region | | | | | | |
| **Kwai Tsing** | M | 470.27 | 666.82 | 599.27 | 115.91 | 608.27 |
| | SD | 101.26 | 79.38 | 64.02 | 4.18 | 31.96 |
| **Lantau** | M | 131.00 | 176.73 | 125.64 | 45.27 | 270.91 |
| | SD | 21.44 | 30.95 | 23.07 | 1.19 | 12.49 |
| **Sha Tin** | M | 477.82 | 845.55 | 525.00 | 147.73 | 707.36 |
| | SD | 109.42 | 74.51 | 55.90 | 1.19 | 1.91 |
| **Tsuen Wan** | M | 329.64 | 520.55 | 461.45 | 118.82 | 518.82 |
| | SD | 81.42 | 49.92 | 85.10 | 3.12 | 29.65 |

**Table 3. Pearson correlations between key variables.**

| | No. detected violent crimes (o1) | No. detected property crimes (o2) | No. detected other crimes (o3) | No. of detectives (i1) | No. of uniformed police (i2) | Triad influence (triad) | Estimated population (pop) | Proportion of youth population (youthpop) |
|---|---|---|---|---|---|---|---|---|
| o1 | 1.0000 | | | | | | | |
| o2 | 0.7685* | 1.0000 | | | | | | |
| o3 | 0.6579* | 0.7714* | 1.0000 | | | | | |
| i1 | 0.6340* | 0.8194* | 0.8354* | 1.0000 | | | | |
| i2 | 0.5005* | 0.4822* | 0.4647* | 0.6624* | 1.0000 | | | |
| triad | 0.6198* | 0.6534* | 0.7082* | 0.7283* | 0.2932* | 1.0000 | | |
| pop | 0.5395* | 0.4517* | 0.0417 | 0.2952* | 0.3687* | 0.2393^ | 1.0000 | |
| youthpop | 0.5239* | 0.1872 | 0.0880 | 0.0335 | 0.0306 | 0.2262^ | 0.4133* | 1.0000 |

*$p < 0.001$

^$p < 0.05$

detections across all crime types. Our results also mostly align with our above discussed expectations (method section) where the higher the district population, the higher the number of detected crimes except for "other" crimes. We also find a positive correlation between number of police and the size of population. Finally, the proportion of youth population in a district is positively correlated with the number of detected violent crimes, triad influence and estimated size of population.

### 5.3 Research question 2

The average overall (DEA-CCR model), technical (DEA-BCC model) and scale efficiency score (CCR/BCC) across districts was 69.7 (SD = 17.6), 74.6 (SD = 16.7), and 93.4 (SD = 11.1), respectively. The DEA findings reveal that 98 percent of the 198 DMUs were found to be overall inefficient. Approximately 93.4 percent of DMUs were found to be technically inefficient and 98 percent of the 198 DMUs were scale inefficient. Tests for returns-to-scale (RTS), based on the data-driven cross-validation bandwidth and 100 bootstrap replications [77], in the output-oriented DEA models rejected the null hypothesis of a constant RTS (bandwidth value = 0.088, $p$ = 0.01) and a non-increasing RTS (bandwidth value = 0.068, $p$ = 0.01). This suggests that the production functioning exhibits an increasing RTS. DMUs with an increasing RTS indicate that detection of crime increases by more than a proportional change in the number of detectives and uniformed police officers after adjusting for triad influence, the size of population, as well as exogenous variables (proportion of youth population and year of measurement) in a district. Results from the DEA-CCR model indicate that only four of the 198 DMUs had an overall efficiency rating of 100 percent–the highest efficiency score achievable. These include Eastern (2009), Mongkok (2007, 2008) and Yuen Long (2007). These DMUs were also found to be technically and scale efficient. Other DMUs that are considered technically efficient include Central (2007, 2016), Lantau (2013, 2015), Mongkok (2016), Wanchai (2012), Western (2007), and Yau Tsim (2008, 2016) (see S2 Appendix for DEA estimates of each DMU).

Our study finds that the overall crime detection inefficiencies are due to scale inefficiencies rather than pure technical inefficiencies. Therefore, police districts can make improvements by adjusting their scales, except those districts that are scale efficient. Table 4 compares districts by their mean efficiency scores and variance for the 11 years.

**Table 4. Mean efficiency score and variance by district.**

| District | DEA-CCR efficiency | | DEA-BCC efficiency | | Scale efficiency | |
|---|---|---|---|---|---|---|
| | Mean | SD | Mean | SD | Mean | SD |
| Central | 71.8 | 9.6 | 87.0 | 8.9 | 82.3 | 4.7 |
| Eastern | 85.3 | 10.8 | 86.1 | 11.2 | 99.1 | 1.2 |
| Kowloon City | 53.3 | 6.8 | 53.7 | 6.8 | 99.4 | 0.2 |
| Kwai Tsing | 68.8 | 10.9 | 71.0 | 12.9 | 97.3 | 2.5 |
| Kwun Tong | 74.8 | 9.7 | 79.6 | 9.1 | 93.8 | 1.7 |
| Lantau | 29.6 | 3.4 | 59.1 | 25.0 | 56.0 | 16.4 |
| Mongkok | 88.8 | 8.7 | 89.9 | 9.2 | 98.9 | 1.3 |
| Sau Mau Ping | 53.7 | 8.3 | 56.9 | 10.4 | 94.7 | 3.3 |
| Sha Tin | 79.9 | 8.8 | 80.9 | 8.6 | 98.7 | 0.6 |
| Sham Shui Po | 74.7 | 7.7 | 75.4 | 7.3 | 99.0 | 0.9 |
| Tai Po | 77.3 | 12.9 | 78.8 | 13.2 | 98.0 | 1.3 |
| Tsuen Wan | 49.4 | 4.8 | 50.9 | 4.6 | 97.0 | 1.1 |
| Tuen Mun | 69.0 | 9.4 | 70.1 | 8.6 | 98.3 | 1.8 |
| Wanchai | 85.2 | 7.0 | 90.8 | 7.4 | 93.8 | 1.0 |
| Western | 63.2 | 9.4 | 74.8 | 12.0 | 84.7 | 3.7 |
| Wong Tai Sin | 60.0 | 9.2 | 61.7 | 10.1 | 97.4 | 1.4 |
| Yau Tsim | 84.7 | 9.8 | 88.6 | 10.5 | 95.7 | 2.1 |
| Yuen Long | 85.6 | 9.5 | 87.6 | 7.5 | 97.6 | 3.0 |

The result of the bias-corrected bootstrap of police efficiency in the detection of crime with 3000 bootstrap replicates at 95% confidence interval is shown in Fig 1. The figure shows that the efficiency score of police districts differs only slightly when corrected for sampling variations. Overall, the average change across efficiency scores for the 198 DMUs are less than 4% for the CCR model and 5% for the BCC model. The four aforementioned scale-efficient police districts become less efficient when corrected for sampling bias, where their efficiency levels drop during the bootstrapping process (see Fig 2). When comparing the ranking of police districts before and after bootstrapping, the rankings in the DEA-CCR model changed by, on average, 4 ranks, and in the DEA-BCC model by about 6 ranks. The rank of most districts remains stable. This is consistent with other DEA studies using bootstrapping [e.g. 78], where the upper-bound results coincide with the original technical efficiency estimates.

## 5.4 Research question 3

Results from the slack analysis demonstrate in what areas and by how much an inefficient police district could improve their allocation of resources (see S3 Appendix). Slacks in the detection of other crimes, violent crime and property crimes were found in 118, 104, and 24 DMUs respectively. This reveals that police districts can improve their detection of violent, property and other crimes on average by 22.1%, 3.7% and 24.5%, respectively. This is calculated by the averages of potential improvement from the original outputs to the recommended target outputs.

We also find that districts with a higher number of detected crimes in all three crime categories appear to have higher overall, technical and scale efficiency (see Table 5). Further, districts with a higher number of detected crimes tend to have higher slacks in the number of detectives and lower slacks in the number of uniformed police. We find a significant and positive correlation between the potential improvement for violent crime and the bias-corrected overall ($r = .338$, $p < .001$) and bias-corrected technical ($r = .299$, $p < .01$) efficiency scores as

(a) Confidence intervals and DEA-CCR bootstrap efficiency

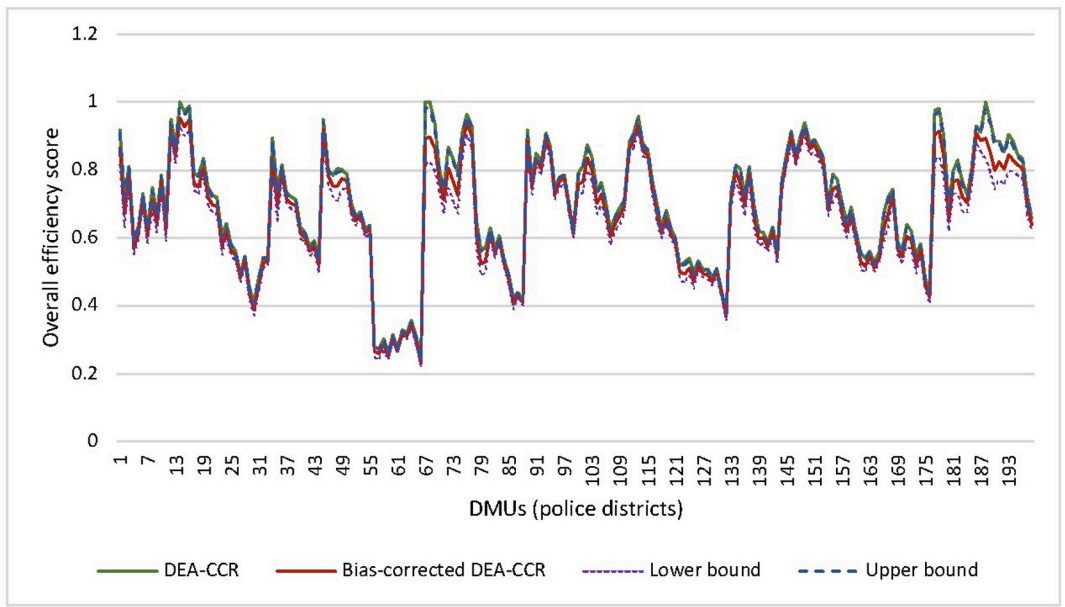

(b) Confidence intervals and DEA-BCC bootstrap efficiency

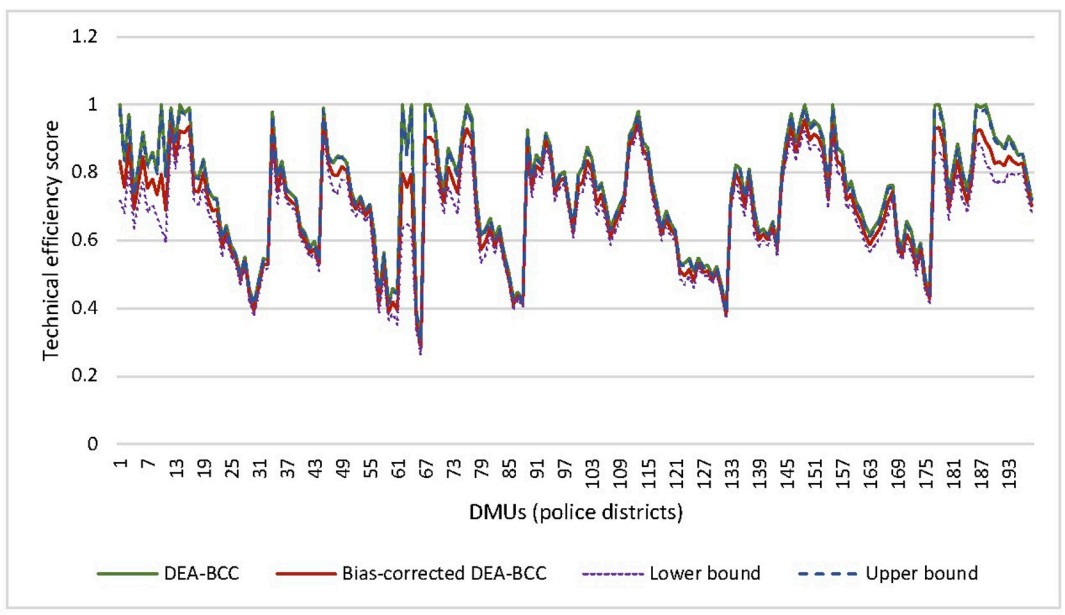

**Fig 2. Bootstrap DEA technical efficiency estimates.**

well as BCC efficiency ($r = .397$, $p < .001$) and scale efficiency ($r = .356$, $p < .001$). BCC slacks on the detection of other crimes are only positively and significantly correlated with the CCR bias-corrected ($r = .274$, $p < .05$) and BCC bias-corrected ($r = .291$, $p < .05$) efficiency but not the original CCR and BCC efficiency scores.

**Table 5. Correlations between district crime rates, efficiency scores, slacks, inputs and outputs.**

| | No. detected violent crimes (o1) | No. detected property crimes (o2) | No. detected other crimes (o3) | No. of detectives[1] (i1) | No. of uniformed police[1] (i2) | CCR efficiency (CCR) | CCR Bias-corrected (CCR-BC) | BCC efficiency (BCC) | BCC Bias-corrected (BCC-BC) | Scale efficiency (Scale) | Slacks in the detection of violent crime (Slacks-V) | Slacks in the detection of property crime (Slacks-P) | Slacks in the detection of other crime (Slacks-O) | Slacks in the number of detective (Slacks-D) | Slacks in the number of uniform police (Slacks-U) |
|---|---|---|---|---|---|---|---|---|---|---|---|---|---|---|---|
| o1 | 0.7685* | 1 | | | | | | | | | | | | | |
| o2 | 0.6579* | 0.7714* | 1 | | | | | | | | | | | | |
| o3 | 0.8613* | 0.9365* | 0.9158* | 1 | | | | | | | | | | | |
| i1 | 0.6139* | 0.7097* | 0.6771* | 0.7398* | 1 | | | | | | | | | | |
| i2 | 0.6200* | 0.6839* | 0.4086* | 0.6151* | 0.8158* | 1 | | | | | | | | | |
| CCR | 0.6019* | 0.8318* | 0.7092* | 0.7979* | 0.2925^ | 0.3089^ | 1 | | | | | | | | |
| CCR-BC | 0.5751* | 0.8312* | 0.6719* | 0.7740* | 0.2841^ | 0.3178* | 0.9958* | 1 | | | | | | | |
| BCC | 0.3965* | 0.5852* | 0.5124* | 0.5597* | -0.0274 | -0.0284 | 0.8662* | 0.8613* | 1 | | | | | | |
| BCC-BC | 0.4501* | 0.6637* | 0.5419* | 0.6179* | 0.0523 | 0.0763 | 0.9081* | 0.9098* | 0.9863* | 1 | | | | | |
| Scale | 0.5263* | 0.6551* | 0.5379* | 0.6339* | 0.6752* | 0.7134* | 0.4933* | 0.4965* | 0.0137 | 0.1127 | | | | | |
| Slacks-V | -0.1599 | 0.3322* | 0.0951 | 0.1263 | 0.147 | 0.0992 | 0.3382* | 0.3971* | 0.2989^ | 0.3559* | 0.1757 | 1 | | | |
| Slacks-P | -0.0632 | -0.248 | 0.143 | -0.0522 | -0.253++ | -0.3440* | 0.1392 | 0.1123 | 0.2478 | 0.2049 | -0.1623 | -0.1173 | 1 | | |
| Slacks-O | 0.1573 | 0.2761^ | -0.2383 | 0.0479 | 0.0418 | 0.2949^ | 0.2301 | 0.274++ | 0.2232 | 0.2907^ | 0.0582 | 0.3927* | -0.1661 | 1 | |
| Slacks-D | 0.1669 | 0.3776* | 0.4383* | 0.3831* | 0.5183* | 0.2990^ | 0.231 | 0.2378 | 0.2066 | 0.2316 | 0.0944 | 0.4055* | 0.0678 | -0.0097 | 1 |
| Slacks-U | -0.277++ | -0.3794* | -0.3645* | -0.3824* | -0.7017* | -0.4432* | 0.0381 | 0.0537 | 0.2528++ | 0.2197 | -0.3592* | 0.1159 | 0.3844* | 0.2232 | -0.2238 |

\*$p < .001$

^$p < .01$

++$p < .05$

Notes: [1] moderated by the triad influence, total no. of population and the proportion of youth in the population

## 6. Discussion

The overall performance of districts was assessed by setting their detection of three categories of crime against the instrument-adjusted manpower levels. We began our analysis by examining the correlation between triad influence, the detection of crime and police strength. We found empirical support that perceived triad influence is positively correlated with the number of detectives and uniformed police, as well as the detection of violent, property and other crimes. We then undertook as series of diagnostic tests for endogeneity and identified instruments to partial out endogeneity. The use of both triad influence and the size of the population as IVs allowed us to adjust our inputs to perform our bootstrap DEA and slack analysis. Comparing results with previous studies is inappropriate given differences in intrinsically different empirical, theoretical and methodological approaches. In addition, the policing approaches also differ across regions resulting in unmeaningful comparisons (e.g., comparing Hong Kong policing with the UK). Locally, while we have undertaken an earlier study using DEA [51], readers should be cautious into making direct comparisons for three reasons: (1) the current study adopts instrument-adjusted inputs for the DEA; (2) the current DEA model adopts a longer reference period (i.e. a change in a peer benchmark DMU would modify the relative efficiency score for a DMU); and (3) the current DEA study uses a different approach to quantify the inputs and outputs and uses DEA-CCR and DEA-BCC models rather than the Free Disposable Hull approach. We see the current study as an improvement in the modelling given access to more relevant data including triad influence across districts.

DEA results identified variation in performance by district regarding detection and their position on the PPF compared to those districts identified as efficient. The district of

Mongkok, with the highest mean overall efficiency score (88.8), was found to be among those districts that have a low variance across years, indicating a more stable performance in crime detection. This is important considering that entertainment businesses in Mongkok are triad-dominated territories [22]. To avoid triad-related disorder, the police deploy their best police into such districts [32].

Previous research [e.g. 62] suggests that low mean efficiency scores tend to be accompanied by high variances across years. In this study, Central, Kwai Tsing and Western appear to follow this pattern. However, districts such as Lantau and Sau Mau Ping were found to produce consistently low mean scores and low variance in overall and technical efficiencies across years. The results derived from our bootstrap DEA demonstrate that the efficiency score of police districts differs only slightly compared to traditional DEA results. The sampling bias, therefore, is not a great concern but we recommend that the bootstrap method becomes a standard in DEA to ensure that any underlying bias is identified.

From here, we were then able to examine what modifications can be made to current levels of detection that would render inefficient districts more efficient (slack analysis). To be technically efficient, the inefficient DMUs could theoretically increase their detection of crimes by adopting more efficient policing models or cross-district police collaboration and joint actions with reference to the resource allocation decisions made by their benchmark peers.

Focusing firstly on the potential improvements on inputs (i.e. the number of detectives and uniform police), the range of improvement is negligible (no more than 5%). However, as discussed above, our output-oriented model focuses on how much output can be proportionally increased without making substantial changes to inputs. As expected, the recommended number of detectives and uniformed police remain relatively stable across the study period and as such there is little potential improvement observed.

Second, improvements on outputs were recommended for all three crime categories, with other crimes and violent crime found to be the areas with more DMUs with slacks compared to property crimes. This result reveals significant improvements in the detection of crime can be made by targeting violent and other crimes in districts such as Kwun Tong and Sha Tin. It is noted that the recommended number of detected crimes across all three crime categories are almost always less than the number of reported crimes in the districts. We have (198DMUs × 3 crime types) 594 cells on outputs and there are 20 occasions when the DEA model suggests detection beyond the number of reported crimes for a given category. Here, the potential for DMUs to detect crime beyond reported crime and hence moving toward their theoretical benchmark is by acknowledging that: (i) there are uncaptured crimes that are not reported to the police in all DMUs; and (ii) the recommended direction of improvement in crime detection (e.g. a given district could detect more violent crime) would hold (i.e. overall efficiency would improve if more violent crimes were detected) even if the DMU might not be able to reach its theoretical and optimal benchmark.

The above inefficient districts would be best placed to examine how the district of Mongkok achieves consistently high levels of detection across all crime types, albeit periods where some improvements can still be made. For the benefit of the reader, Mongkok is situated in Kowloon and is recognised as a major tourist shopping area. It has a high population density and is often portrayed in films as well as perceived by residents as an area run by triads given the large number of bars, nightclubs and massage parlours.

Our results reveal that Mongkok reports a higher-than-average detection in property and other crimes. The district was efficient at the start of the reference period (i.e. 2007–2008), however there was a noticeable drop in the number of detected property and other crimes after 2009, with only minor changes to police strength. We also note that the number of violent, property and other crimes being reported to police in Mongkok did not change

substantially across the reference period except the final two years. When compared to other districts which tend to have comparable detections across violent, property and other crime types (with property and other crime detections being approximately 30% more than the detection of violent crime), Mongkok reveals a disproportionate detection of crime across the three crime categories–where the detection of violent crime is consistently and substantially lower than that of property and other crimes (by at least 50 percent). It should be noted that Mongkok, compared to most other districts, reveals consistent efficiency despite the aforementioned limitations. The reasons for the disproportionate detection of crime by categories are twofold. First, like other night-time entertainment districts around the world, Mongkok historically has had its share of alcohol-related violence and crime, including triad-related crime [79]. The transient nature of the population who frequent the district makes police investigations into these crimes potentially challenging. Second, triads who actively engaged in Mongkok tend to have their own factions of junior gang members who commit crimes in Mongkok but reside in other districts often evading police detection [34]. This suggests that cross-district police collaboration is essential to increase police efficiency in tackling triad-related crime.

Our results from the slack analysis of Mongkok also reveal potential inefficiencies that could be ratified by the reallocation of human resources during inefficient years. For example, in 2011, had the police district been operating efficiently Mongkok could have theoretically detected more violent crime (by approximately 29% or 131 additional arrests). Being a district with high triad penetration (as captured by our instrument-adjusted DEA), Mongkok police commanders would begin by questioning why the detection of property and other crimes is higher than violent crimes. Here, it may appear that police officers have particular skills that are amenable to the detection of property and other crimes. The question for the commanders is whether these skills are easily translated to the detection of violent crime or if personnel with such skills are needed to be brought in from other districts or if additional training can be provided to existing officers.

The results of our DEA are important as official statistics would reveal that the detection of violent crime in proportion to reported crime is not particularly low. It is only through the DEA that we can identify where inefficiency lies when compared to other districts. Without DEA, police commanders would be under the impression that no improvement is necessary when in fact the number of arrests could be theoretically increased. This would not be unique to Hong Kong and thus additional studies that assist police in making empirically informed decisions are necessary. Here, we note that future research could employ alternative qualitative methods (e.g. text mining techniques and latent Dirichlet allocation) to provide additional insights into resource allocation decisions.

The use of DEA regarding the detection of crime and police resource allocation is still in its infancy where there are opportunities to use DEA to assess efficiency in other jurisdictions. Much could be learnt from the contextual variation that most likely exist between jurisdictions and how different types of organised crime groups (e.g. Russian Mafia) influence crime dynamics and how DEA can better inform resource allocation decisions that disrupt organised crime. Opportunities also exist to extend this analysis by including an evaluation of expenditure and funding allocation at the district level. This information has the potential to improve decisions regarding deployment of resources on police strength, police composition, and method of policing such as foot patrols and undercover policing. Beyond resource allocation, DEA also provide a technique for: (1) improving accountability and transparency–the method provides a data-driven approach to evaluate the performance of police activities; (2) performance improvement–identifying inefficiencies and areas where resources may not be optimally utilised, police can implement targeted improvements, leading to a more effective and

responsive police force; (3) evidence-based policymaking–DEA provides valuable evidence for policymaking related to law enforcement, where results inform decisions about resource allocation, staffing, and strategies for crime reduction; (4) stakeholder engagement–DEA can foster better communication and engagement with the community, as citizens have access to data-driven assessments of law enforcement agencies; (5) resource optimisation–DEA can help police optimise their resource allocation, ensuring that limited resources are used efficiently to achieve desired outcomes, such as crime reduction and public safety; (6) cost savings–identifying inefficiencies and areas for improvement can potentially lead to cost savings for law enforcement agencies, allowing them to reallocate resources to other critical areas or invest in community policing initiatives; and (7) ongoing adaptation to changing contextual variation–frontier models assist police in making data-driven decisions that are sensitive to changing circumstances (e.g. change in triad influence across districts). In summary, the method we use in this paper can lead to improved efficiency, transparency, and policymaking in law enforcement, ultimately resulting in better public safety outcomes.

## Supporting information

**S1 Appendix. Data envelopment analysis (DEA) models on policing services.**
(PDF)

**S2 Appendix. Window analysis of efficiency in crime detection across all districts.**
(PDF)

**S3 Appendix. DEA-BCC slack results for each input and output for the DMUs.**
(PDF)

**S1 File.**
(XLSX)

## Acknowledgments

We would like to thank Professor Wing Lo for providing the data on the perceived triad influence across Hong Kong Districts. We would also like to thank Dr Sharon Kwok for meeting with us at the early stages of the project and providing her informative knowledge on triad history in Hong Kong.

## Author Contributions

**Conceptualization:** Gabriel Wong, Matthew Manning, T. Wing Lo.

**Data curation:** Gabriel Wong, Matthew Manning, T. Wing Lo.

**Formal analysis:** Gabriel Wong, Matthew Manning, T. Wing Lo.

**Investigation:** Gabriel Wong, Matthew Manning, T. Wing Lo.

**Methodology:** Gabriel Wong, Matthew Manning, T. Wing Lo.

**Project administration:** Matthew Manning.

**Supervision:** Matthew Manning.

**Writing – original draft:** Gabriel Wong, Matthew Manning.

**Writing – review & editing:** Matthew Manning, T. Wing Lo, Shane D. Johnson.

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
