## [Decision Letter · Decision Letter 0]

12 Sep 2023

PONE-D-23-15806Triad Influence on the Detection of Crime in Hong KongPLOS ONE

Dear Dr. Johnson,

Thank you for submitting your manuscript to PLOS ONE. After careful consideration, we feel that it has merit but does not fully meet PLOS ONE’s publication criteria as it currently stands. Therefore, we invite you to submit a revised version of the manuscript that addresses the points raised during the review process.

We look forward to receiving your revised manuscript.

Kind regards,

Adetayo Olorunlana, Ph.D.

Academic Editor

PLOS ONE

Journal Requirements:

Reviewers' comments:

Reviewer's Responses to Questions

**Comments to the Author**

1. Is the manuscript technically sound, and do the data support the conclusions?

Reviewer #1: Yes

Reviewer #2: Yes

2. Has the statistical analysis been performed appropriately and rigorously? 

Reviewer #1: Yes

Reviewer #2: Yes

3. Have the authors made all data underlying the findings in their manuscript fully available?

Reviewer #1: Yes

Reviewer #2: Yes

4. Is the manuscript presented in an intelligible fashion and written in standard English?

Reviewer #1: Yes

Reviewer #2: Yes

5. Review Comments to the Author

Reviewer #1: Dear authors

Initially, I would like to congratulate you on your research proposal. The topic is relevant because it deals with the application of science to identify local crime rates, in this case in Hong Kong, the search for an understanding of the phenomena surrounding criminal actions and the search for police strategies that act to reduce crime, this has a direct impact on improving people's quality of life, which is why I consider the research to be relevant. The authors used DEA as the main methodology of the work: the description of the method is adequate and clear. I would only suggest that they describe all the method's formulas. The results are consistent, the discussion is satisfactory. The conclusion is based on the results. The work has some gaps to be filled in or explained by the authors, which I will describe below:

1) I suggest that in the abstract and conclusion the authors indicate the practical and social implications of the research, as well as indicating future research;

2) The authors used the DEA method; the DEA method is recurrent in this topic. There are other methodologies that also seek to understand the phenomenon of crime, I suggest that the authors insert a short paragraph informing readers that the topic studied also has other biases, such as multi-criteria methods, topical modeling, text mining, etc. Then present the justifications for the choice of DEA, I think it would be more consistent. In this sense, I suggest considering the following papers: DOI: 10.13189/ujm.2017.051202 (Review of the Literature on Multicriteria Methods Applied in the Field of Public Security); https://doi.org/10.1108/DTA-12-2018-0109 (Identification of operational demand in law enforcement agencies: An application based on a probabilistic model of topics).

3) The authors have structured the Introduction in subsections, which have replaced the literature review. In this sense, I suggest that the authors introduce a subsection that provides a short theoretical overview of the evolution of policing strategies developed and applied in various realities. I believe that this summary is important for readers' understanding, and then, as the authors have done, introduce the reality studied in Hong Kong. To do so, I suggest reading and considering the following paper: https://doi.org/10.1108/JM2-10-2020-0268 (Knowledge discovery in research on policing strategies: an overview of the past fifty years);

4) Number all the equations;

5) review the formatting of the references introduced in the text and in the references section. Check the authors' instructions;

6) Insert the numbering and title of the figures at the bottom of the figure; also insert the period in the title of the figures;

Good proofreading

Reviewer #2: Thanks for the opportunity to review this work.

1. The title of this project is well defined and timely in that Clime.

2. The methodology involved in the analysis is really innovative and appropriate to the study.

3. The answers to the research questions were properly enumerated except for the third research question " how can the allocation of policing resources be adjusted to improve crime detection?" which requires a qualitative research design to approach it for appropriate and logical solution, not quantitative approach as used in this study.

4. The table 5 in the discussion section should be removed and placed in the right section.

5. The discussion aspect should employed more literature for comparison and robust analysis.

6. PLOS authors have the option to publish the peer review history of their article (what does this mean?). If published, this will include your full peer review and any attached files.

Reviewer #1: **Yes: **Dr. Marcio Basilio

Reviewer #2: No

---

## [Author Response · Author response to Decision Letter 0]

7 Dec 2023

Response to Reviewers

Reviewer #1 

Overall comment:

Dear authors

Initially, I would like to congratulate you on your research proposal. The topic is relevant because it deals with the application of science to identify local crime rates, in this case in Hong Kong, the search for an understanding of the phenomena surrounding criminal actions and the search for police strategies that act to reduce crime, this has a direct impact on improving people's quality of life, which is why I consider the research to be relevant. The authors used DEA as the main methodology of the work: the description of the method is adequate and clear. I would only suggest that they describe all the method's formulas. The results are consistent, the discussion is satisfactory. The conclusion is based on the results. 

Authors’ response: 

Thank you for the comments. We have addressed all the constructive points in below.

Comment 1: The work has some gaps to be filled in or explained by the authors, which I will describe below:

1) I suggest that in the abstract and conclusion the authors indicate the practical and social implications of the research, as well as indicating future research;

Authors’ response: We have added additional information in the abstract and conclusion consistent with your recommendation.

Comment 2: The authors used the DEA method; the DEA method is recurrent in this topic. There are other methodologies that also seek to understand the phenomenon of crime, I suggest that the authors insert a short paragraph informing readers that the topic studied also has other biases, such as multi-criteria methods, topical modeling, text mining, etc. Then present the justifications for the choice of DEA, I think it would be more consistent. In this sense, I suggest considering the following papers: DOI: 10.13189/ujm.2017.051202 (Review of the Literature on Multicriteria Methods Applied in the Field of Public Security); https://doi.org/10.1108/DTA-12-2018-0109(Identification of operational demand in law enforcement agencies: An application based on a probabilistic model of topics).

Authors’ response: We have added a section to discuss this as proposed by the reviewer.

Comment 3: The authors have structured the Introduction in subsections, which have replaced the literature review. In this sense, I suggest that the authors introduce a subsection that provides a short theoretical overview of the evolution of policing strategies developed and applied in various realities. I believe that this summary is important for readers' understanding, and then, as the authors have done, introduce the reality studied in Hong Kong. To do so, I suggest reading and considering the following paper: https://doi.org/10.1108/JM2-10-2020-0268 (Knowledge discovery in research on policing strategies: an overview of the past fifty years);

Authors’ response: We have added a new section “ 1.1.1 A brief history of Hong Kong policing strategies” to address this comment.

Comment 4: Number all the equations;

Authors’ response: Equations are now all numbered.

Comment 5: review the formatting of the references introduced in the text and in the references section. Check the authors' instructions;

Authors’ response: We have updated formatting as requested.

Comment 6: Insert the numbering and title of the figures at the bottom of the figure; also insert the period in the title of the figures;

Authors’ response: Changes have been made according to the guideline.

 

Reviewer #2: Thanks for the opportunity to review this work.

Comment 1. The title of this project is well defined and timely in that Clime.

Authors’ response: Thanks.

Comment 2. The methodology involved in the analysis is really innovative and appropriate to the study.

Authors’ response: Thanks.

Comment 3. The answers to the research questions were properly enumerated except for the third research question " how can the allocation of policing resources be adjusted to improve crime detection?" which requires a qualitative research design to approach it for appropriate and logical solution, not quantitative approach as used in this study.

Authors’ response: We have addressed this specific research question at the best capacity possible with the DEA model. Frontier modelling is designed to specifically address this question of resource allocation. The field of frontier analysis is well established, and the modelling has been applied across many different area to address this research question on resource allocation improvement in quantitative methods. We acknowledge that an alternative model can be qualitative by nature, but the DEA model is still capable of answering this question as discussed in the results and discussion sections. We have also added a new sentence in the discussion to highlight that future research could employ alternative qualitative methods to supplement our quantitative approach.

Comment 4. The table 5 in the discussion section should be removed and placed in the right section.

Authors’ response: Thank you for the comment. We have made this change.

Comment 5. The discussion aspect should employed more literature for comparison and robust analysis.

Authors’ response: We have added a few sentences in the first paragraph of the discussion to highlight reasons for not making comparisons with previous studies where there are significant differences for four reasons: (1) the current study adopts instrument-adjusted inputs for the DEA; (2) the current DEA model adopts a longer reference period when compared to previous local studies (i.e. a change in 1 peer benchmark DMU would change the relative efficiency score in the equation); (3) the current DEA study uses a different approach to quantify the inputs and outputs and uses DEA-CCR and DEA-BCC models rather than the Free Disposable Hull approach; and (4) jurisdiction differences due to different background and approaches in policing strategies.

---

## [Decision Letter · Decision Letter 1]

29 Dec 2023

Triad Influence on the Detection of Crime in Hong Kong

PONE-D-23-15806R1

Dear Dr. Johnson,

We’re pleased to inform you that your manuscript has been judged scientifically suitable for publication and will be formally accepted for publication once it meets all outstanding technical requirements.

Kind regards,

Adetayo Olorunlana, Ph.D.

Academic Editor

PLOS ONE

Additional Editor Comments (optional):

Reviewers' comments:

Reviewer's Responses to Questions

**Comments to the Author**

1. If the authors have adequately addressed your comments raised in a previous round of review and you feel that this manuscript is now acceptable for publication, you may indicate that here to bypass the “Comments to the Author” section, enter your conflict of interest statement in the “Confidential to Editor” section, and submit your "Accept" recommendation.

Reviewer #1: All comments have been addressed

Reviewer #2: All comments have been addressed

2. Is the manuscript technically sound, and do the data support the conclusions?

Reviewer #1: Yes

Reviewer #2: Yes

3. Has the statistical analysis been performed appropriately and rigorously? 

Reviewer #1: N/A

Reviewer #2: Yes

4. Have the authors made all data underlying the findings in their manuscript fully available?

Reviewer #1: Yes

Reviewer #2: Yes

5. Is the manuscript presented in an intelligible fashion and written in standard English?

Reviewer #1: Yes

Reviewer #2: Yes

6. Review Comments to the Author

Reviewer #1: Dear Authors

I congratulate you on the extensive revision of the current text. I note that you have taken on board all the suggestions made by the reviewers in the first round of revision. After analyzing the current version, I found no areas for improvement that would substantially increase the quality of the current text. I am therefore considering this text for publication.

Best regards

Reviewer #2: Thanks for making the necessary adjustments to this paper. The concerns raised initially in the methodology and discussion sections have ben adequately addressed. Hence, making the work appropriate for publication.

7. PLOS authors have the option to publish the peer review history of their article (what does this mean?). If published, this will include your full peer review and any attached files.

Reviewer #1: **Yes: **Dr. Marcio Basilio

Reviewer #2: **Yes: **Oluseye Ademola Okunola

---

## [Editor Report · Acceptance letter]

7 Feb 2024

PONE-D-23-15806R1 

PLOS ONE

Dear Dr. Johnson, 

I'm pleased to inform you that your manuscript has been deemed suitable for publication in PLOS ONE. Congratulations! Your manuscript is now being handed over to our production team.

Kind regards, 

on behalf of

Associate Professor Adetayo Olorunlana 

Academic Editor

PLOS ONE